# Molybdenum and Nickel Nanoparticles Synthesis by Laser Ablation towards the Preparation of a Hydrodesulfurization Catalyst

**Viviana Londoño-Calderón [1]** , **Rogelio Ospina [2]** , **Jhonatan Rodriguez-Pereira [2]**,
**Sergio A. Rincón-Ortiz [2] and Elisabeth Restrepo-Parra [1],\***

[1]  Laboratorio de Física del Plasma, Universidad Nacional de Colombia—Manizales, Km. 9 vía al Aeropuerto, Campus La Nubia, Manizales 170001, Colombia; vilondonoca@unal.edu.co

[2]  Centro de Investigación Científica y Tecnológica en Materiales y Nanociencias (CMN), Universidad Industrial de Santander, Piedecuesta, Santander 681011, Colombia; ROSPINAO@uis.edu.co (R.O.); Jhonatan.RodriguezPereira@upce.cz (J.R.-P.); contab@uis.edu.co (S.A.R.-O.)

\*  Correspondence: erestrepopa@unal.edu.co; Tel./Fax: +57-(6)-887-9495

**Abstract:** A clean straightforward laser ablation method in deionized (DI) water is reported for the synthesis of Molybdenum (Mo) and Nickel (Ni) nanoparticles (NPs). The structural, morphological, and optical properties of the as-synthesized nanoparticles were investigated. Particle size was estimated to be less than 10 nm, the UV–vis spectra of the samples show the formation of $H_2MoO_4$ and NiO. The XRD results for the Ni sample show the presence of two phases, cubic nickel oxide, and an fcc metallic nickel phase, indicating the possible formation of Ni/NiO compound. The nanoparticles synthesized were used as precursors in the production of a NiMo/$\gamma$-$Al_2O_3$ catalyst. The textural and structural properties, chemical composition, and catalytic performance in a hydrodesulfurization (HDS) reaction are reported. The textural and structural properties results show the lack of pore-blocking due to the small sizes and the distribution of the metallic nanoparticles on the support. Chemical composition measured by XPS shows a ratio Ni/Mo of 1.34. Therefore, possibly Ni was deposited on Mo covering part of its active area, occupying active sites of Mo, removing its effective surface and resulting in a relatively low conversion of DBT (17%). A lower Ni/Mo ratio is required to improve the model system, which could be achieved by changing parameters at the production of the nanoparticles. The model system can also be further tuned by changing the size of the nanoparticles.

**Keywords:** laser ablation; nickel; molybdenum; catalyst; HDS

## 1. Introduction

Currently, the oil industry faces a challenge that requires attention, the high content of heavy molecules with heteroatoms of sulfur, nitrogen, metals, and oxygen on crude oils. The removal of these pollutant compounds from oil-derived products is possible using hydrotreating (HDT) under the processes of hydrodesulfurization (HDS) [1], hydrodenitrogenation (HDN) [2], hydrodemetallization (HDM) [3], and hydrodeoxygenation (HDO) [4] processes in which refractory molecules react with hydrogen in the presence of a catalyst, under conditions of high pressure and temperature, to remove heteroatoms of sulfur, nitrogen, metals, and oxygen, respectively [5]. Among these compounds, sulfur is present in higher concentrations in oil-derived products, the removal of this element is important for different reasons: environmental regulations related to sulfur and aromatic content are constantly increasing regarding the regulations on the number of toxic substances allowed in fuels [6,7]. On the other hand, sulfur corrodes pipes and poisons the catalysts used in post-refining processes [8].

On the other hand, many transition metals have been used as catalysts for HDS reactions like $RuS_2$ [9], Au-Pd/$SiO_2$ [10], and $Rh_2S_3$ [11]. However, it is not widely used in the industry due to its high fabrication cost. In the case of supported catalysts based on carbides and transition metal nitrides, it has also shown properties similar to those of noble metals in HDT reactions [12]. However, the industrial application of this kind of catalyst remains difficult due to the instability of the active carbide or nitride phases which may undergo sulfidation very easily, even in the presence of low sulfur concentrations. Mo or W sulfides promoted by Ni or Co supported on gamma-alumina ($\gamma$-$Al_2O_3$) are usually preferred for industrial uses due to its high activity [13]. Mo or W are responsible for catalytic activity, Ni or Co increase catalytic activity in terms of reaction rate per site, this set of compounds is known as the active phase and the support is regularly $\gamma$-$Al_2O_3$ due to its high surface area that stabilizes the dispersion of the active component [14]. The catalytic performance is directly related to the dispersion of the active species on the support, for this reason, an important factor in the coherent production of HDT catalysts is the particle size of the nanoparticle catalysts [15]. Additionally, the size and shape of a nanoparticle determine the structure and chemical composition of a nanoparticle, and these parameters influence the activity and selectivity of the catalyst. The metals for the production of this type of catalyst have been synthesized using different chemical methods. For example, Liu et al. [16] fabricated an unsupported NiMo catalyst, the precursors were synthesized by PVP-assisted chemical precipitation. The authors studied the effect of polyvinylpyrrolidone (PVP) on the catalytic activity of NiMo catalysts. The results showed that a NiMo catalyst synthesized with 0.5 g of PVP has a higher DBT conversion than the other NiMo catalysts. Additionally, Scott et al. [17] prepared NiMo sulfide nanoparticles by precipitation employing water-in-oil microemulsions. The authors obtained 3–5 nm nanoparticles that produced a 50% HDS conversion of a vacuum gas oil (VGO) as well as improved HDN activity compared to a conventional NiMo/$\gamma$-$Al_2O_3$ catalyst. However, chemical synthesis generally involves the production of contaminating residues, in addition to significant energy consumption that generates $CO_2$ emissions [18]. Lately, the chemical industry emphasizes the importance of developing more sustainable production processes and therefore the importance of sustainable nanoparticle synthesis. Laser ablation in liquids represents a simple, low-abrasion, and low-contaminant material synthesis technique compared to other techniques, that obey the principles of green chemistry and benefits the cost of the materials because no molecular precursors are required for laser ablation synthesis the reagent costs are lower. Therefore, laser ablation in water is proposed as an alternative to the use of chemical reagents. The technique has not been widely explored in the synthesis of HDS catalyst nanoparticles and allows the production of numerous species with controlled morphologies and homogeneous sizes [19]. In this paper, the synthesis of Ni and Mo catalyst nanoparticles by laser ablation in liquids, evaluating the morphological, optical, and structural properties of the NPs synthesized through Atomic Force Microscopy (AFM), ultraviolet–visible spectroscopy (UV–vis), and X-ray diffraction (XRD), and the fabrication of an NiMo supported catalyst, evaluating the textural, structural, and compositional properties of the catalyst by the use of nitrogen adsorption/desorption, X-ray diffraction (XRD), and X-ray photoelectron spectroscopy (XPS), as well as its catalytic performance in a hydrodesulfurization reaction, is reported.

## 2. Results

### 2.1. Nanoparticles

The average particle size was studied through DLS measurements; for each sample, the mean size and one standard deviation were calculated, the results show an average size of 20.36 ± 3.67 nm for Mo sample and 14.53 ± 3.51 nm for Ni sample. Particle size was also determined using AFM technique. Figure 1 shows AFM micrographs of Ni and Mo nanoparticles distributed on a silica substrate. The images reveal quasi-spherical morphologies for both samples. The average sizes of NPs were determined by measuring at least one hundred objects and creating a histogram. For each data set, the mean size and one standard deviation were calculated, the results show an average size of

4.82 ± 2.22 nm for Mo sample and 6.02 ± 2.10 nm for Ni sample. The difference in particle size from DLS and AFM techniques is related to the specificity of each technique. AFM measures the geometric size of the NPs and DLS measures the hydrodynamic diameter of the theoretical nanoparticle with the same diffusion coefficient as the measured nanoparticle. As a result, the size of a nanoparticle measured by DLS can differ from the one determined by AFM.

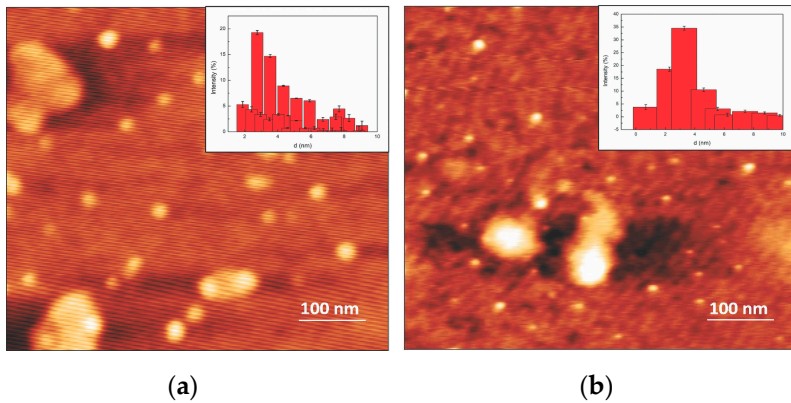

|  |  |
|:---:|:---:|
| (**a**) | (**b**) |

**Figure 1.** AFM images and histograms of (**a**) Mo nanoparticles (NPs), (**b**) Ni NPs.

Figure 2 shows the UV–vis absorption spectrum for (a) Mo and (b) Ni samples. The broad band in the range of 207–235 nm corresponds to the tetrahedral Mo species $(MoO_4)^{2-}$, when the surface of Mo particles reacts with water then the molybdate ion appears in solution. The dissolution reaction generates protons, and as a result, the water becomes acidic. This behavior was reported by Ozeki et al. [20] from the UV–vis spectrum of three mononuclear species: $(MoO_4)^{2-}$, $(HMoO_4)^{-}$, and $(H_2MoO_4)$. It is important to note that molybdate $(MoO_4)^{2-}$ is tetrahedral, whereas the polymolybdates and molybdenum trioxide are octahedral. Therefore, octahedral structures usually show additional and broader peaks, that are shifted toward greater wavelengths, compared with tetrahedral structures. It can be deduced then, that Mo samples synthesized can be attributed to the species $H_2MoO_4$, the formation of this species is mostly present in acidic environments according to the literature [21]. This can be associated with the high value of laser energy, a larger concentration of nanoparticles is generated in these conditions and therefore, a greater number of free protons that interact with the liquid medium are produced causing the liquid medium to become acidic. The UV–vis curve was fitted with Lorentzian functions. The intensity values of the peaks are 3.57 for the peak located at 204.25 nm and 3.59 for the peak located at 249.72 nm; the similitudes between the intensity values of both peaks indicate the protonation of the molybdate anion. UV–vis spectra of nickel nanoparticles in water (Figure 2b) exhibit a peak located at 252 nm characteristic from NiO nanoparticles formation [22]. The bandgap values of Ni and Mo samples were estimated employing the Tauc method [23], by plotting $(\alpha h\nu)^{1/2}$ as a function of energy $(h\nu)$ and extrapolating the linear region to the abscissa (where $\alpha$ is the adsorption coefficient, h is Planck's constant, and $\nu$ is the vibration frequency), as shown in the inset of Figure 2a. The reported bulk bandgap for $H_2MoO_4$ is 3.18 eV and the calculated bandgap value was 4.05 eV. In the case of Ni sample, the value calculated was of 4.38 eV; this value is higher than the one reported in the literature for bulk NiO (4.0 eV) [24]. Nanoparticles usually exhibit a higher bandgap than the bulk value. Bulk materials are formed by a large number of atoms and molecules and therefore a fusion of many adjacent energy levels. As the particle size reaches the nanometric scale, where each particle is made of a very small number of atoms or molecules, the number of orbitals overlapping or energy levels decreases, and the bandwidth is reduced. This leads to an increase in the gap between the valence band and the conduction band [25].

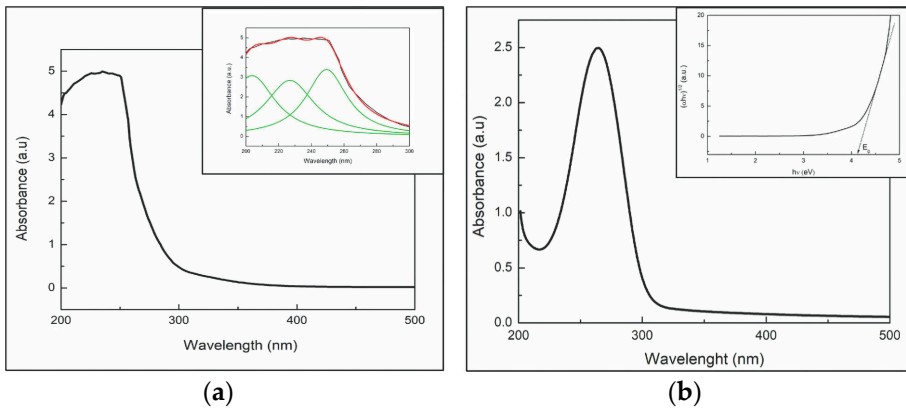

**Figure 2.** UV–vis spectrum and Tauc method for (**a**) Mo NPs and (**b**) Ni NPs.

XRD pattern of the produced Ni NPs is shown in Figure 3. The results show a peak placed at $2\theta = 44.55°$, attributed to Si (220) [26], and associated with the silicon substrates from the sample preparation. The additional peaks were identified as Ni (111), Ni (200), Ni (012), Ni (222), NiO (111), NiO (012), and NiO (220) indicating the formation of a pure cubic nickel oxide phase (bunsenite, NaCl structure) [27], and an fcc metallic nickel phase [28]. Small Ni particles can be produced in the early stages of particle formation and can grow by the accumulation of several of those particles. These larger particles can suffer surface oxidation, due to the partial decomposition of water under laser interaction, producing Ni/NiO structures. The crystallite size was calculated for the sample using the Scherrer formula [29]. The value obtained was 5.84 nm, which means the particles were formed by smaller crystallites. In the case of Mo NPs, the results above show that the formed compound when molybdenum came in contact with water was molybdic acid ($H_2MoO_4$), other molybdenum-based compounds can be formed during the drying process for the preparation of the sample for XRD measurement. Therefore, a quantitative analysis of the crystalline structure is not possible. According to the literature, there is no direct evidence of the crystalline structures of these materials, since most structural methods require high concentrations where conversion to other molybdenum-based compounds occurs [30].

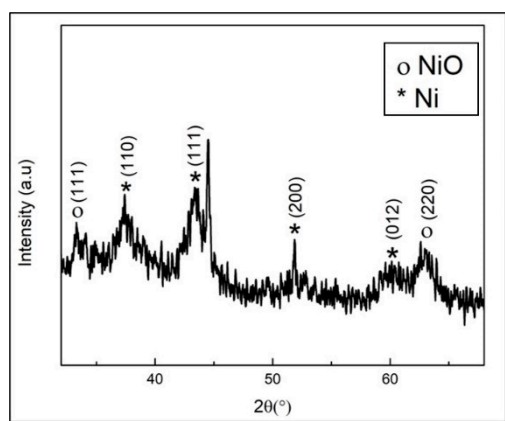

**Figure 3.** XRD pattern for Ni NPs.

### 2.2. NiMo/γ-Al₂O₃ Catalyst

Table 1 shows the textural properties of the support and the catalyst in different stages like specific surface area ($S_{BET}$), total pore volume ($V_p$), and the average pore diameter ($D_p$). According to Table 1 and taking alumina as a reference, it can be observed that there are no significant changes in the values of specific surface area, pore volume, and pore diameter with the incorporation of the Mo and Ni metals.

**Table 1.** Textural properties.

|  | $S_{BET}$ (m$^2$/g) | $V_P$ (cm$^3$/g) | $D_P$ (nm) |
|---|---|---|---|
| $\gamma$-Al$_2$O$_3$ | 236.79 | 0.70 | 8.98 |
| Mo/$\gamma$-Al$_2$O$_3$ | 227.04 | 0.62 | 8.57 |
| NiMo/$\gamma$-Al$_2$O$_3$ | 235.55 | 0.61 | 8.51 |

$S_{BET}$ = Specific surface area; $V_P$ = Pore volume; $D_P$ = Average pore diameter.

Based on the classification adopted by the International Union of Applied Chemistry (IUPAC) for the types of adsorption and hysteresis isotherms, it is deduced from Figure 4 that the solids evaluated have type IV (a) isotherms, which correspond to mesoporous materials (20–500 Å), and H2 (b) type hysteresis, corresponding to materials with complex porosity since the pores have a neck-type geometry in which the molecular transit through them is tortuous [31]. All the samples presented similar adsorption–desorption isotherms.

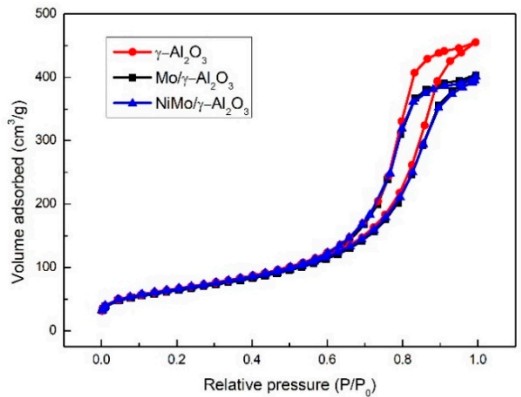

**Figure 4.** Adsorption and hysteresis isotherms on different stages of the catalyst fabrication.

The XRD patterns of fresh and spent catalysts are shown in Figure 5. For both cases, no detectable XRD diffraction peaks of any Mo and Ni oxide species were observed. Notably, no changes in the bulk of the catalyst were indicated from the powder X-ray diffraction patterns of the materials, which revealed only diffractions peaks consistent with $\gamma$-Al$_2$O$_3$ at 2$\theta$ of 19.45, 32.02, 37.74, 39.48, 45.91, 60.89, and 66.95° [32]. In addition, the signal intensity appeared to be lower in the spent catalyst. This behavior is usually attributed to the formation of an uneven top-layer with the island of metallic nanoparticles generated on the support surface after the sulfide process on the catalyst [33].

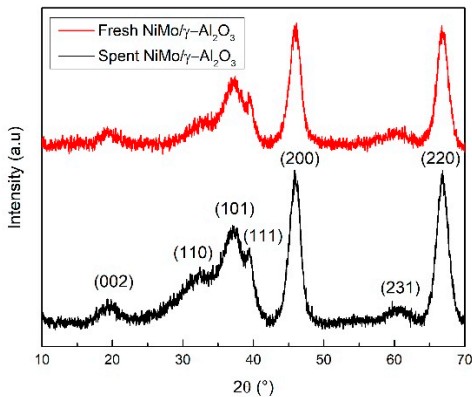

**Figure 5.** XRD pattern for fresh and spend NiMo/$\gamma$-Al$_2$O$_3$.

Figure 6 shows the results of the high-resolution XPS spectra of the precursor metals on the NiMo/γ-Al$_2$O$_3$ catalyst. The two peaks at 856.40 and 873.50 eV in the Ni 2p XPS spectra as shown in Figure 6a are attributed to the spin-orbit split lines of Ni 2p$_{3/2}$ and Ni 2p$_{1/2}$. The two peaks around 862.08 and 879.58 eV correspond to the satellite structures of Ni 2p$_{3/2}$ and Ni 2p$_{1/2}$ [34]. The Ni compounds in the sample may be attributed to NiO, Ni$_2$O$_3$, spinel NiMoO$_4$ as a result of the interaction between NiO and MoO$_3$, or spinel NiAl$_2$O$_4$ as a result of the interaction of NiO with Al$_2$O$_3$ [35]. However, the nonexistence of a peak within the nominal range of 853.5–854.5 eV for Ni 2p$_{3/2}$ suggests the absence of NiO on the sample. The Ni 2p$_{3/2}$ binding energy of 856.40 eV is in the nominal range of 855.5–857.5 for Ni$_2$O$_3$, but according to the literature [34,35], the appearance of an almost symmetric peak at 856.40 eV and a shakeup satellite structure at 862.08 eV proposes the existence only of Ni$^{2+}$ in the surface. Consequently, the Ni peaks shown in Figure 6a can be assigned to Ni$^{2+}$ in NiAl$_2$O$_4$ or NiMoO$_4$. On the other hand, the decomposition of the Mo 3d$_{3/2}$ and 3d$_{5/2}$ high resolution XPS spectra of catalyst are shown in Figure 6b, the results show the existence of two different types of Mo species present on the surface. The peaks at 233.36 and 236.49 eV are assigned to Mo$^{6+}$ in MoO$_3$ [34] or NiMoO$_4$ [35]. In addition, two shoulder peaks at 231.94 and 235.07 eV correspond to Mo$^{4+}$ in MoO$_2$ [34].

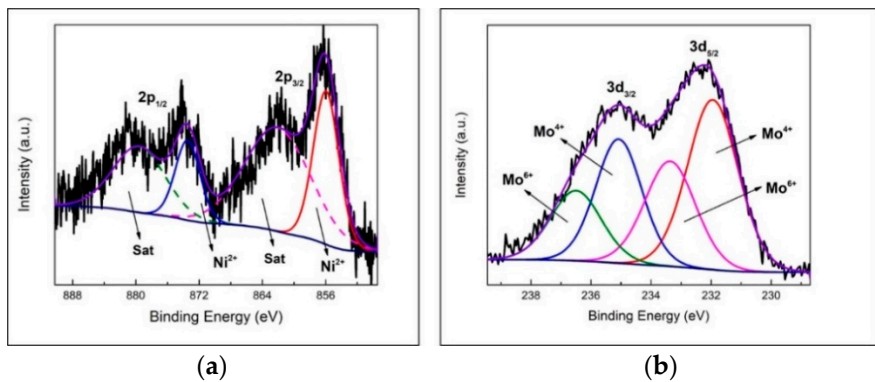

|           (a)           |           (b)           |

**Figure 6.** High-resolution XPS spectra of (**a**) Mo 3d, (**b**) Ni 2p.

The Ni and Mo content were investigated using XPS. Table 2 lists the atomic percentage of Ni and Mo as well as the ratio between the metals on the catalyst. The results of Ni/Al and Mo/Al ratios indicate a preferable accumulation of Ni species than Mo species on the surface.

**Table 2.** XPS surface composition.

| | Surface Composition | | |
|---|---|---|---|
| | **Ni/Mo** | **Ni/Al** | **Mo/Al** |
| NiMo/γ-Al$_2$O$_3$ | 1.34 | 0.11 | 0.08 |

The catalytic activity of sulfide NiMo/γ-Al$_2$O$_3$ catalyst was examined in the hydrodesulfurization of DBT. The conversion and yield percentages for NiMo/γ-Al$_2$O$_3$ catalyst calculated are shown in Figure 7a. Results show that DBT was transformed into cliclohexylbenzene (CHB) and biphenyl (BF), as products of hydrogenation (HYD) and direct desulfurization (DDS) routes respectively, while potential intermediaries of the HYD pathway, tetrahydro-dibenzothiophene (THDBT) were presented in a very low percentage (1.2%). Furthermore, no other intermediate products were found. In the case of conversion, it can be seen that the DBT presents a conversion of around 17%. On the other hand, Figure 7b shows the selectivity of the catalyst, where the NiMo/γ-Al$_2$O$_3$ catalyst presents a selectivity to DDS since it presents a higher yield to biphenyl. This behavior is typical and occurs for commercial catalysts of this type as reported by Morales et al. [36].

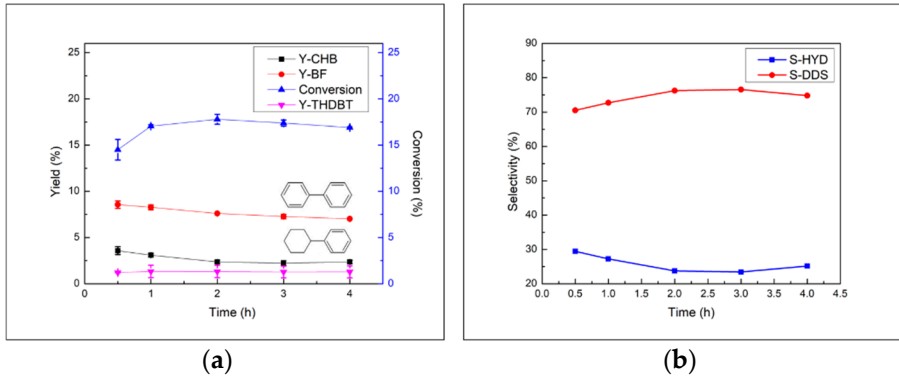

**Figure 7.** Catalytic performance of NiMo/γ-Al$_2$O$_3$ catalyst. (**a**) Conversion and yield. (**b**) Selectivity.

## 3. Discussion

An industrial catalyst is formed of irregularly shaped nanoparticles in a wide variety of sizes. In the case of a bimetal catalyst, an important variable is the possibly different composition of each metallic nanoparticle of the catalyst. The catalytic performance of an industrial catalyst is the sum of the contributions of each nanoparticle with structural and chemical variations. However, due to the interactions between structural and chemical factors, it is quite difficult to gain fundamental knowledge of how each of these factors influences the catalytic performance. Moreover, fundamental knowledge is crucial in the coherent design of a catalyst with high activity, selectivity, and durability. Changes in chemical and structural parameters of catalyst nanoparticles, such as size and shape, generally vary the chemistry and surface structure of nanoparticles, and consequently the selectivity and catalytic activity of the catalyst. All chemical processes involved in heterogeneous catalysis are performed on the surface of the catalyst nanoparticles. Therefore, it is essential to establish an intrinsic correlation between different properties of catalyst nanoparticles and the catalytic performance of the catalyst. According to the results of the textural properties presented in Table 1, no significant changes in the values of surface area and pore volume were noticed as the metals were impregnated on the surface of the alumina. Generally, the increase in the size of the crystallite of nanoparticles generates pore blockage reducing the specific surface area and pore volume, probably the small sizes of the metallic particles favor the dispersion of the active phase in the support without generating agglomerations and pore-blocking on the support surface. This behavior can be verified by XRD results were no Mo oxides or Ni oxides species were observed indicating the lack of agglomerations on the support surface. In addition, Mo and Ni oxides are considered to be evenly distributed on the surface of the support. On the other hand, the presence of both Mo$^{6+}$ and Mo$^{4+}$ electronic structures on the surface of the catalyst suggests an uneven distribution of Mo in the electronic environment on the surface. During impregnation and drying, small crystallites of the catalytic precursor are deposited on the inner surface of the support. These steps involve mass and/or heat transfer processes that frequently do not reach equilibrium, resulting in non-uniform concentration profiles throughout the support. The total catalytic precursor deposited on the support surface for interaction impregnation comes from two sources: the first is the solute adsorbed during the impregnation step and the second comes from the non-adsorbed solute that precipitates during the drying stage [37]. Additionally, the formation of the Ni$^{2+}$, Mo$^{6+}$, and Mo$^{4+}$ species on the surface of the catalyst, serves as a precursor of the Ni-Mo-S active phase due to the interactions that exist between the Ni-Mo-O species. The quantification shows an atomic ratio Ni/Mo value of 1.34, indicating enrichment of Ni in the surface, and a higher amount of Ni atoms in the surface than Mo atoms. The Ni/Mo ratio is highly relevant for the activation of the catalyst, and the average atomic percentage of the active metals. A Ni/Mo ratio of around 0.5 is desired for applications in hydrodesulfurization reactions, the fabricated catalyst exhibit a ratio considerably higher indicating an excess of Ni in the system. This excess can influence catalytic performance in two ways: (i) promoting the formation of other nickel-based compounds on the surface of the support, such as Ni$_3$S$_2$ or NiS.

The compounds $Ni_3S_2/NiS$ does not play a catalytic role in HDS reactions, despite its amount or degree of dispersion [38] or (ii) promoting the substitution of Mo by Ni, generating the loss of NiMoS active sites, resulting in a relatively low catalyst conversion rate. A lower Ni/Mo ratio is required to improve the model system, which could be achieved by changing parameters at the production of the nanoparticles. The model system can be also be further tuned by changing the size of the nanoparticles.

## 4. Materials and Methods

### 4.1. Nanoparticles

High purity Mo (99.9%) and Ni (99.9%) solid targets in DI water were ablated using a Q-smart 850 Nd: YAG Quantel laser (pulse duration 8 ns, repetition rate 10 Hz) using the 532 nm line of the laser source. The Mo target was exposed in 25 mL for the laser energy of 300 mJ with an ablation time of 10 min. While the Ni target was exposed in 10 mL to laser energies of 150 mJ with an ablation time of 7 min. Laser-generated nanoparticles were characterized by employing an Atomic Force Microscope (AFM) measurements were performed with a commercial AFM system (HITACHI 5100N), operating under ambient conditions. Images were obtained in the tapping mode using a self-sensitive micro cantilever PRC-DF40P. To carry out AFM measurements, Mo and Ni nanoparticle colloids were deposited on commercial silicon substrates using the procedure described elsewhere [39]. UV–vis absorption spectroscopy in the spectral range of $\lambda$ = 200–800 nm was done using a SHIMADZU model UV-2401 PC, in a quartz cuvette with a 1 mm path length. The structural characterization was performed using an X-ray diffractometer (XRD) D8 Bruker AXS Diffractometer, composed of an X-ray source of Cu K$\alpha$, with $\lambda$ = 1.5406 Å and a secondary monochromator of graphite.

### 4.2. NiMo/γ-Al$_2$O$_3$ Catalyst

Supported NiMo catalysts were prepared by wetness impregnation of Ni and Mo nanoparticles previously synthesized by laser ablation on commercial Procatalyse $\gamma$-$Al_2O_3$ support. The powders of $\gamma$-$Al_2O_3$ were macerated and then sieved to get the desired size (0.3–0.6 mm). The loadings were Ni oxide (1.8 wt%) and Mo (8 wt%), the loadings were calculated as the loss of mass on each target. The Mo precursor was impregnated first, after each impregnation the materials were subjected to static rest at room temperature for 8 h to achieve the diffusion of the metallic nanoparticles through the pores of the support. Afterward, the materials were rotary evaporated in a Buschi rotary evaporator to separate the solute from the solvent. After each impregnation step, the samples were dried at 120 °C for 12 h, and calcined at 500 °C in air for 4 h. The textural properties, such as specific surface area (SBET), pore volume (VP), and average pore diameter (DP) were characterized by nitrogen adsorption–desorption isotherms at 77 K using a 3FLEXTM (Micromeritics). SBET values were calculated using constant minimization (CBET) [40]. $V_P$ and $D_P$ values were calculated using the Barret, Joyner, and Halenda (BJH) [41]. Before measurements, the samples were degassed under vacuum at 393 K for 1 h and subsequently at 573 K for 12 h. Under these conditions, a final pressure of 0.05 mbar is always required. The weighed amount of each material for the tests was about 0.15 g. The structural characterization was carried out using an X-ray diffractometer (XRD) D8 Bruker AXS Diffractometer, composed of an X-ray source of Cu K$\alpha$, with $\lambda$ = 1.5406 Å with a register range from 2 to 70° and a step size of 0.02035°. The chemical composition was characterized through X-ray photoelectron spectroscopy with a SPECS spectrometer, equipped with a PHOIBOS 150 2D-DLD energy analyzer. The spectra were recorded using monochromatic Al K$\alpha$ radiation (h$\nu$ = 1486.6 eV) operated at 100 W. Spectra were processed and analyzed with the CASA–XPS software using the relative sensitivity factors (R.S.F) provided by the manufacturer. The binding energy of adventitious carbon, C-(C, H) at 284.8 eV [42] was used as a reference to adjust the binding energy scale of the spectra.

### 4.3. Catalytic Tests

The DBT HDS reaction was conducted at a temperature of 400 °C under a pressure of 5 MPa, with a liquid charge flow of 30 mL/h and an $H_2$ flow/liquid charge ratio of 500 LN/L, a time on stream of 9 h, and a weight hour space velocity (WHSV) of 52.54 $h^{-1}$. The catalytic activity was evaluated using a load consisting of 2 wt% dimethyldibenzothiophene (DBT) as a model molecule, 96 wt% cyclohexane as the solvent, and 2 wt% hexadecane as an internal standard for chromatography. The reactions were carried out in a continuously flowing tubular fixed reactor loaded with 0.15 g catalyst diluted with inert sand to keep a constant volume of 2 mL. Before each reaction, the catalyst was dried in situ at 120 °C under a flow of $N_2$ for 1 h, with a heating ramp of 10 °C/min. At the end of this stage, an activation was performed with a mixture of $H_2S$ and $H_2$ (15–85%) at 400 °C for 4 h, with a heating ramp of 10 °C/min. The gas flow was 100 mL/min in both the drying and activation processes. During each catalytic test, the condensable reaction products were collected every hour. The condensable products were analyzed by gas chromatography on an HP 6890 GC equipment equipped with an FID detector and HP-1 column (100 m × 25 mm × 0.5 µm) and a split injector. A scheme of the complete experimental process is shown in Figure 8.

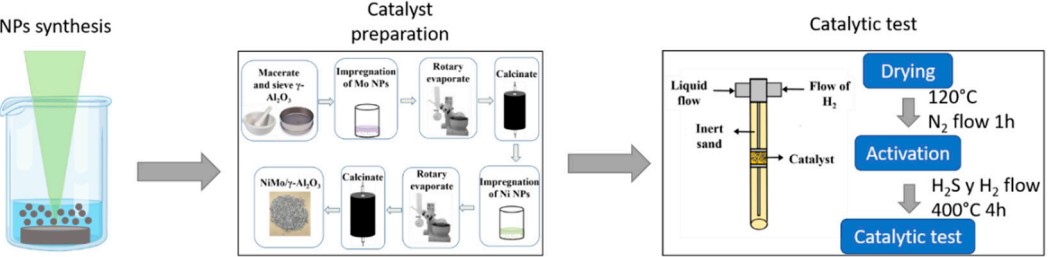

**Figure 8.** Experimental design.

## 5. Conclusions

Ni and Mo oxides nanoparticles were prepared using laser ablation in water and they were later used as precursors on the fabrication of a NiMo catalyst. The obtained nanoparticles were characterized by using AFM, UV–vis, and XRD. Results from the AFM technique showed that the average particle size for both samples was smaller than 10 nm. UV–vis results indicated the formation of ($H_2MoO_4$) and NiO. XRD diffractograms indicated the presence of two phases: cubic nickel oxide and fcc metallic nickel. The textural, compositional, and structural properties of the catalyst were characterized. Nitrogen Adsorption/desorption and XRD measurements show the lack of pore-blocking on the sample due to the small sizes of the nanoparticles. XPS results showed the existence of the species $Ni^{2+}$, $Mo^{6+}$, $Mo^{4+}$ on the surface of the catalyst, these species serve as precursors of the Ni-Mo-S active phase due to the interactions that exist between the Ni-Mo-O species. Results from the HDS reaction of DBT show a conversion percentage of 17% and a selectivity to the DDS route. This behavior can be attributed to the high ration Ni/Mo indicating an Ni excess on the catalyst. Therefore, possibly Ni was deposited on Mo covering part of the active area of it occupying active sites of Mo, removing its effective surface and resulting in a relatively low conversion of DBT.

**Author Contributions:** Conceptualization, V.L.-C., J.R.-P., and R.O.; methodology, R.O. and J.R.-P.; formal analysis, V.L.-C. and S.A.R.-O.; investigation, V.L.-C., J.R.-P., and S.A.R.-O.; resources, E.R.-P. and R.O.; writing—original draft preparation, V.L.-C.; writing—review and editing, E.R.-P.; supervision, E.R.-P. All authors have read and agreed to the published version of the manuscript.

**Funding:** This research received no external funding.

**Conflicts of Interest:** The authors declare no conflict of interest.

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
