# Peer review of "Molybdenum and Nickel Nanoparticles Synthesis by Laser Ablation towards the Preparation of a Hydrodesulfurization Catalyst"

_catalysts, doi:10.3390/catal10091076_

Round 1
Reviewer 1 Report
The present manuscript discusses the formation of molybdenum and nickel oxides nanoparticles of less than 10 nm in size using a laser ablation method. The prepared materials deposited on alumina and then tested in the HDS reaction for DBT. The manuscript could be published after major revision and addressing the following comments:1) Please add the reaction conditions (temperature, pressure, time on stream, GHSV, etc.) under which the HDS reactions were conducted to Fig 7.
2) Authors claimed the particle size of less than 10 nm based on the AFM images while there are large particles above 10 or even 20 nm in the provided images which left wit no clarification or discussion. TEM images should be provided to support the size of nanoparticles. It is also interesting if the authors compare the size of nanoparticles prepared via the conventional impregnation method.
3) Please clarify how the loadings of 1.8% and 8 % for Ni and Mo were determined. There is no information about the actual metal contents in the bulk catalysts, which should be measured using either XRF or atomic absorption or ICP methods. Thus, the conclusion drawn from XPS in Table 2 about the surface concentration of Ni-Mo is not correct, considering that several factors could contribute to such deviation such as Ni and Mo particle size difference. On the other hand, it seems based on BET data that the synthesized particles are mostly present on the surface and XPS is strong enough for scanning the top 8 nm of the surface.
4) The formation of bimetallic Ni-Mo structures was not verified by either EDX, XRD, or XPS, as no shifts in the Mo or Ni XPS binding energies were observed in XPS. Given that, a conclusion about the surface segregation of Ni is not correct.
5) Considering the above comment, what is the advantage of utilizing this synthesis method compared to the conventional impregnation on the HDS reaction, as the effects of size or structure were not studied? It was reported that a Ni/Mo ratio of 0.5 is desired for HDS; what was the reason that samples with much higher Ni/Mo were prepared from the first place? Thus, it would be very interesting to prepare a sample of corresponding Mo and Ni contents of the present sample using impregnation and then tested under similar HDS conditions, or a sample with Ni/Mo of 0.5 be prepared using the presented synthesis method. Is it also possible to prepare bimetallic Ni-Mo nanoparticles in a single solution?
6) Please indicates the types of materials precursors for alumina and Ni and Mo and their concentration in the solution for laser ablation synthesis.
Author Response
The present manuscript discusses the formation of molybdenum and nickel oxides nanoparticles of less than 10 nm in size using a laser ablation method. The prepared materials deposited on alumina and then tested in the HDS reaction for DBT. The manuscript could be published after major revision and addressing the following comments:
- Please add the reaction conditions (temperature, pressure, time on stream, GHSV, etc.) under which the HDS reactions were conducted to Fig 7.
Response: The reaction conditions have been clarified on the materials and method section on the following paragraph:
The DBT HDS reaction was conducted at a temperature of 400°C under a pressure of 5 MPa, with a liquid charge flow of 30 ml/h and an H2 flow/liquid charge ratio of 500 LN/L, a time on stream of 9 h, and a WHSV of 52.54 h-1.
- Authors claimed the particle size of less than 10 nm based on the AFM images while there are large particles above 10 or even 20 nm in the provided images which left with no clarification or discussion. TEM images should be provided to support the size of nanoparticles. It is also interesting if the authors compare the size of nanoparticles prepared via the conventional impregnation method.
Response: The authors’ intended to determine the average particle size of isolated/individual particles, larger particles are attributed to aggregate particles. Unfortunately, we do not have the sources to perform TEM measurements, this was the main reason to use the AFM technique to determine the average particle size of the nanoparticles. We include DLS measurements to support the information determined with AFM and an explication of the difference on the value of the sizes determined by both techniques has been provide in the following sentences:
The average particle size was studied through DLS measurements, for each sample, the mean size and one standard deviation were calculated, the results show an average size of 20.36 ± 3.67 nm for Mo sample and 14.53 ± 3.51 nm for Ni sample.
The difference in particle size from DLS and AFM techniques is related to the specificity of each technique. AFM technique measures the geometric size of the NPs and DLS measures the hydrodynamic diameter of the theoretical nanoparticle with the same diffusion coefficient as the measured nanoparticle. As a result, the size of a nanoparticle measured by DLS can differ from the one determined by AFM.
- Please clarify how the loadings of 1.8% and 8 % for Ni and Mo were determined. There is no information about the actual metal contents in the bulk catalysts, which should be measured using either XRF or atomic absorption or ICP methods. Thus, the conclusion drawn from XPS in Table 2 about the surface concentration of Ni-Mo is not correct, considering that several factors could contribute to such deviation such as Ni and Mo particle size difference. On the other hand, it seems based on BET data that the synthesized particles are mostly present on the surface and XPS is strong enough for scanning the top 8 nm of the surface.
Response: The loadings of 1.8wt% and 8wt% were the theoretical values impregnated to the alumina, these values were selected according to literature for this type of catalyst. Measurements to determine the actual metal contents in the bulk catalysts could not be performed. Therefore, the loadings on the catalyst may differ from the theoretical values and consequently, the Ni-Mo ratio may differ as well. Also, different factors related to the physics properties of the nanoparticles and the synthesis method may affect the Ni-Mo ratio.
- The formation of bimetallic Ni-Mo structures was not verified by either EDX, XRD, or XPS, as no shifts in the Mo or Ni XPS binding energies were observed in XPS. Given that, a conclusion about the surface segregation of Ni is not correct.
Response: The phrase has been removed from the text; we thank the reviewer for the observation
- Considering the above comment, what is the advantage of utilizing this synthesis method compared to the conventional impregnation on the HDS reaction, as the effects of size or structure were not studied? It was reported that a Ni/Mo ratio of 0.5 is desired for HDS; what was the reason that samples with much higher Ni/Mo were prepared from the first place? Thus, it would be very interesting to prepare a sample of corresponding Mo and Ni contents of the present sample using impregnation and then tested under similar HDS conditions, or a sample with Ni/Mo of 0.5 be prepared using the presented synthesis method. Is it also possible to prepare bimetallic Ni-Mo nanoparticles in a single solution?
Response: Laser ablation in liquids was proposed as an alternative to the use of pollutant chemical precursors in the fabrication of HDS nanoparticle catalysts. A paragraph expanding the advantages of the technique has been added to the introduction in the following paragraph:
Lately, the chemical industry emphasizes the importance of developing more sustainable production processes and therefore the importance of sustainable nanoparticle synthesis. Laser ablation in liquids represents a simple, low-abrasion, and low-contaminant material synthesis technique compared to other techniques, that obey the principles of green chemistry and benefits the cost of the materials because no molecular precursors are required for laser ablation synthesis the reagent costs are lower. Therefore, laser ablation in water is proposed as an alternative to the use of chemical reagents. The technique has not been widely explored in the synthesis of HDS catalyst nanoparticles
- The authors' intention was not produced a Ni/Mo ratio higher than 0.5. The loadings on the catalyst may differ from the theoretical values and consequently, the Ni-Mo ratio may differ as well. Also, different factors related to the physics properties of the nanoparticles and the synthesis method may affect the Ni-Mo ratio.
- We thank the suggestion of the reviewer; our workgroup has a future work with the main to produce a catalyst with a lower Ni/Mo ratio to improve the catalytic performance.
- It is possible to prepare bimetallic Ni-Mo nanoparticles in a single solution by creating a NiMo target and perform laser ablation on it, in the case of this paper the objective was to follow the steps used on the industry to fabricate a NiMo support catalyst, that is the reason to synthetized Mo and Ni nanoparticles separated.
- Please indicates the types of materials precursors for alumina and Ni and Mo and their concentration in the solution for laser ablation synthesis.The precursors for alumina and Ni and Mo and their concentration are shown on the material and method section:
- High purity Mo (99.9%) and Ni (99.9%) solid targets
- Commercial Procatalyse γ-Al2O3 support
- The loadings were Ni oxide (1.8%wt) and Mo (8%wt) and γ-Al2O3 (90.2%wt), the loadings were calculated as the loss of mass on each target
Reviewer 2 Report
The authors investigated in this study the catalytic efficiency of Mo-Ni nanoparticles synthetized by laser ablation in HDS reactions. While the study is certainly of interest to the Journal’s audience, the novelty claimed is limited as there are previous studies on Ni-Mo nanoparticles fabrication by laser ablation and the application of Ni-Mo catalysts in HDS is well reported. Furthermore, some claims made in the manuscript are not convincingly backed up by the characterization conducted. The current version of the manuscript should also be thoroughly reviewed to amend several typos, a complete editing and proofreading by an English native speaker is recommended. For these reasons, the manuscript cannot be accepted in the current version.
The manuscript can be further improved on the basis of the following comments:
- Laser ablation of Ni and Mo nanoparticles is not new, see for instance Marzun G. et al., DOI: 10.1016/j.jcis.2016.09.014. The authors could clarify the novelty statement.
- TEM should be conducted on the synthetized Ni-Mo to support the evidence acquired via AFM. The AFM micrographs are also somehow blurry and the magnification used is quite broad (200 nm).
- “The UV-vis curve was fitted with Lorentzian functions. The intensity values of the peaks are 3.57 for the peak located at 95 25 nm and 3.59 for the peak located at 249.72 nm”. The fitting is not reported in the manuscript, hence this claim cannot be assessed.
- Why are the authors interested in the band gap of the nanoparticles? This information is redundant in the context of this study.
- XPS can identify surface oxides and at%, however to discuss any surface segregation or surface/bulk variation TEM-EDX is required
- There are no repeats or error bars in the study. Experiments should be conducted in triplicate to have some statistical insight.
- Is there any catalyst deactivation/poisoning in the process? Additional repeats on the same substrate could indicate whether the performance deteriorates upon a few cycles.
Author Response
The authors investigated in this study the catalytic efficiency of Mo-Ni nanoparticles synthetized by laser ablation in HDS reactions. While the study is certainly of interest to the Journal’s audience, the novelty claimed is limited as there are previous studies on Ni-Mo nanoparticles fabrication by laser ablation and the application of Ni-Mo catalysts in HDS is well reported. Furthermore, some claims made in the manuscript are not convincingly backed up by the characterization conducted. The current version of the manuscript should also be thoroughly reviewed to amend several typos, a complete editing and proofreading by an English native speaker is recommended. For these reasons, the manuscript cannot be accepted in the current version.
The manuscript can be further improved on the basis of the following comments:
- Laser ablation of Ni and Mo nanoparticles is not new, see for instance Marzun G. et al., DOI: 10.1016/j.jcis.2016.09.014. The authors could clarify the novelty statement.
Response: The novelty statement has been clarified in the introduction on the following sentence
Lately, the chemical industry emphasizes the importance of developing more sustainable production processes and therefore the importance of sustainable nanoparticle synthesis. Laser ablation in liquids represents a simple, low-abrasion, and low-contaminant material synthesis technique compared to other techniques, that obey the principles of green chemistry and benefits the cost of the materials because no molecular precursors are required for laser ablation synthesis the reagent costs are lower. The technique allows the production of numerous species with controlled morphologies and homogeneous sizes [19]. Therefore, laser ablation in water is proposed as an alternative to the use of chemical reagents. The technique has not been widely explored in the synthesis of HDS catalyst nanoparticles
- TEM should be conducted on the synthetized Ni-Mo to support the evidence acquired via AFM. The AFM micrographs are also somehow blurry and the magnification used is quite broad (200 nm).
Response: Unfortunately, we do not have the sources to perform TEM measurements, this was the main reason to use the AFM technique to determine the average particle size of the nanoparticles. The AFM micrographs have been replaced for a 100 nm magnification, also DLS measurements have been provided to support the results presented from AFM and an explication of the difference on the value of the sizes determined by both techniques has been provide in the following sentences:
The average particle size was studied through DLS measurements, for each sample, the mean size and one standard deviation were calculated, the results show an average size of 20.36 ± 3.67 nm for Mo sample and 14.53 ± 3.51 nm for Ni sample.
The difference in particle size from DLS and AFM techniques is related to the specificity of each technique. AFM technique measures the geometric size of the NPs and DLS measures the hydrodynamic diameter of the theoretical nanoparticle with the same diffusion coefficient as the measured nanoparticle. As a result, the size of a nanoparticle measured by DLS can differ from the one determined by AFM.
- “The UV-vis curve was fitted with Lorentzian functions. The intensity values of the peaks are 3.57 for the peak located at 95 25 nm and 3.59 for the peak located at 249.72 nm”. The fitting is not reported in the manuscript; hence this claim cannot be assessed.
Response: The UV-vis fitting curve has been added to the inset of Figure 2.
- Why are the authors interested in the band gap of the nanoparticles? This information is redundant in the context of this study.
Response: The bandgap value is a good approximation to know if the material is nanostructured, this idea is explained in the following sentence:
Nanoparticles usually exhibit a higher bandgap than the bulk value. Bulk materials are formed by a large number of atoms and molecules and therefore a fusion of many adjacent energy levels. As the particle size reaches the nanometric scale, where each particle is made of a very small number of atoms or molecules, the number of orbitals overlapping or energy levels decreases, and the bandwidth is reduced. This leads to an increase in the gap between the valence band and the conduction band.
- XPS can identify surface oxides and at%, however to discuss any surface segregation or surface/bulk variation TEM-EDX is required
Response: The phrase has been removed from the text, we thank the reviewer for the observation
- There are no repeats or error bars in the study. Experiments should be conducted in triplicate to have some statistical insight.
Response: Error bars have been added to Figure 7
- Is there any catalyst deactivation/poisoning in the process? Additional repeats on the same substrate could indicate whether the performance deteriorates upon a few cycles
Response: Our experiments were cut down due to the global pandemic. Therefore, the deactivation/poisoning could not and still cannot be studied.
Reviewer 3 Report
The article is well prepared and it can be interesting for specialists who work in the same discipline. But before publishing, I have some suggestions on how to improve the paper.
There are only two articles from last year. Please add some newest publications to references and describe them in the Introduction section and the novelty of the paper should be better expressed.
Materials and methods section should be before results. Additionally, please add the diagram which shows the process and parameters. It makes the paper more readable.
In the abstract, there should be more concrete results.
Author Response
The article is well prepared and it can be interesting for specialists who work in the same discipline. But before publishing, I have some suggestions on how to improve the paper. There are only two articles from last year. Please add some newest publications to references and describe them in the Introduction section and the novelty of the paper should be better expressed. Materials and methods section should be before results. Additionally, please add the diagram which shows the process and parameters. It makes the paper more readable.
In the abstract, there should be more concrete results.
- Newest publications have been added to references
- T. Sekoai et al., “Application of nanoparticles in biofuels: An overview,” Fuel, vol. 237, pp. 380–397, 2019.
- X. Li et al., “Heterogeneous sulfur-free hydrodeoxygenation catalysts for selectively upgrading the renewable bio-oils to second generation biofuels,” Renew. Sustain. Energy Rev., vol. 82, pp. 3762–3797, 2018.
- D. A. P. Mora and H. Labrador, “efecto de la inserción del benzoato de cobalto en los asfaltenos,” Rev. la Fac. Ciencias, vol. 9, no. 1, pp. 55–71, 2020.
- Liu, W. Li, J. Feng, X. Gao, and Z. Luo, “Promotional effect of TiO2 on quinoline hydrodenitrogenation activity over Pt/γ-Al2O3 catalysts,” Chem. Eng. Sci., vol. 207, pp. 1085–1095, 2019.
- F. Aldosari, “Selective conversion of furfuryl alcohol to 2-methylfuran over nanosilica supported Au: Pd bimetallic catalysts at room temperature,” J. Saudi Chem. Soc., vol. 23, no. 7, pp. 938–946, 2019.
- Liu, S. Jiang, W. Lai, X. Yi, L. Yang, and W. Fang, “Quantitative relationship model between support properties and dibenzothiophene hydrodesulfurization conversion over NiMo/Al2O3,” React. Kinet. Mech. Catal., vol. 121, no. 2, pp. 673–687, 2017.
- The novelty of the paper has been addressed in the following paragraph:
Lately, the chemical industry emphasizes the importance of developing more sustainable production processes and therefore the importance of sustainable nanoparticle synthesis. Laser ablation in liquids represents a simple, low-abrasion, and low-contaminant material synthesis technique compared to other techniques, that obey the principles of green chemistry and benefits the cost of the materials because no molecular precursors are required for laser ablation synthesis the reagent costs are lower. The technique allows the production of numerous species with controlled morphologies and homogeneous sizes [19]. Therefore, laser ablation in water is proposed as an alternative to the use of chemical reagents. The technique has not been widely explored in the synthesis of HDS catalyst nanoparticles
- The order of the manuscript was present according to the template provided by the journal
- A diagram with the process has been added to the paper in Figure 8
- More results have been added to the abstract in the following paragraph:
Chemical composition measured by XPS shows a ratio Ni/Mo of 1.34. Therefore, possibly Ni was deposited on Mo covering part of its active area, occupying active sites of Mo, removing its effective surface and resulting in a relatively low conversion of DBT (17%). A lower Ni/Mo ratio is required to improve the model system, which could be achieved by changing parameters at the production of the nanoparticles. The model system can be also be further tuned by changing the size of the nanoparticles
Round 2
Reviewer 2 Report
I am pleased with the efforts that the authors have put into the current review. It is unfortunate that a part of the characterisation could not be completed due to the impact of covid-19. Nonetheless, as the scientific quality and soundness has been improved, the revised version can be accepted. A minor editing and review of the English style is recommended before the final publication.
Reviewer 3 Report
The authors have improved the article according to my suggestions.
In my opinion, the paper is ready to be published.